# *OsLEA1b* Modulates Starch Biosynthesis at High Temperatures in Rice

**DOI:** 10.3390/plants12234070

**Published:** 2023-12-04

**Authors:** Gang Li, Ruijie Cao, Liuyang Ma, Guiai Jiao, Pengfei Chen, Nannan Dong, Xinwei Li, Yingqing Duan, Xiaoxue Li, Mingdong Zhu, Gaoneng Shao, Zhonghua Sheng, Shikai Hu, Shaoqing Tang, Xiangjin Wei, Yinghong Yu, Peisong Hu

**Affiliations:** 1State Key Laboratory of Rice Biology, China National Center for Rice Improvement, China National Rice Research Institute, Hangzhou 310006, China; Ganglee0102@163.com (G.L.); caoruijie@caas.cn (R.C.); maliuyang2018@163.com (L.M.); jiaoguiai@caas.cn (G.J.); 82101215140@caas.cn (P.C.); 13126531292@163.com (N.D.); lixinwei162013@163.com (X.L.); 18355093180@163.com (Y.D.); xx5990302@126.com (X.L.); shaogaoneng@caas.cn (G.S.); shengzhonghua@caas.cn (Z.S.); hushikai@caas.cn (S.H.); tangshaoqing@caas.cn (S.T.); 2Hunan Rice Research Institute, Changsha 410125, China; uhz_uhz@hotmail.com

**Keywords:** high temperature, *OsLEA1b*, rice grain quality, starch biosynthesis

## Abstract

High temperatures accelerate the accumulation of storage material in seeds, often leading to defects in grain filling. However, the mechanisms regulating grain filling at high temperatures remain unknown. Here, we want to explore the quality factors influenced by the environment and have identified a *LATE EMBROYGENESIS ABUNDANT* gene, *OsLEA1b*, a heat-stress-responsive gene in rice grain filling. *OsLEA1b* is highly expressed in the endosperm, and its coding protein localizes to the nucleus and cytoplasm. Knock-out mutants of *OsLEA1b* had abnormal compound starch granules in endosperm cells and chalky endosperm with significantly decreased grain weight and grain number per panicle. The *oslea1b* mutants exhibited a lower proportion of short starch chains with degrees of polymerization values from 6 to 13 and a higher proportion of chains with degrees from 14 to 48, as well as significantly lower contents of starch, protein, and lipid compared to the wild type. The difference was exacerbated under high temperature conditions. Moreover, *OsLEA1b* was induced by drought stress. The survival rate of *oslea1b* mutants decreased significantly under drought stress treatment, with significant increase in ROS levels. These results indicate that *OsLEA1b* regulates starch biosynthesis and influences rice grain quality, especially under high temperatures. This provides a valuable resource for genetic improvement in rice grain quality.

## 1. Introduction

Rice (*Oryza sativa*) is an important crop consumed as a staple food by more than half of the world’s population. The rice endosperm is the main storage organ for nutrients, including starch, proteins, and lipids [1]. Starch is the main endosperm component, and its content, composition, and structure greatly affect grain quality and yield [2]. Sucrose is efficiently converted into starch through a series of reactions during rice endosperm development [3].There are three main types of enzymes involved in starch biosynthesis. Starch synthase (SS) lengthens the non-reducing end of glucose chains; branching enzymes (BEs) then generate branches from the existing chain; and debranching enzymes (DBEs) hydrolyze branches [4]. In addition, several non-enzymatic proteins participate in starch biosynthesis in the cereal endosperm, such as PROTEIN TARGETING TO STARCH (PTST) and the starch-binding protein FLOURY ENDOSPERM 6 (FLO6) [5,6].

The environment also affects endosperm starch biosynthesis. Heat-induced crop damage is becoming increasingly common as a result of global warming, severely limiting high and stable rice yields [7]. Rice can tolerate certain temperature ranges during different developmental stages [8]. High temperatures perturb the expression of starch biosynthesis-related genes, hindering starch biosynthesis and adversely affecting rice during the grain-filling period [9]. Suppression of *OsMADS7*, encoding a transcription factor, in rice endosperm stabilizes amylose content under high-temperature stress [10]. Inhibiting the expression of α-amylase genes *Amy1A*, *Amy1C*, *Amy3A*, *Amy3D*, and *Amy3E* reduces starch degradation and results in chalky grains under high-temperature conditions [11]. The high pre-mRNA splicing efficiency of *Waxy* (*Wx*) likely stabilizes amylose content in rice seeds under high temperatures [12]. Moreover, genes specifically expressed in the endosperm play an important role in crop growth and development. 

There are several endosperm-specific genes that were identified through a genomic survey [13]. These genes encode grain storage proteins and proteins related to starch biosynthesis, such as the drought-stress-responsive LATE EMBROYGENESIS ABUNDANT (LEA) proteins [14]. In Arabidopsis (*Arabidopsis thaliana*), LEA proteins are associated with seed dehydration tolerance [15]. Four NAC transcription factors bind the *OsLEA* promoter to activate its expression to high levels in embryonic tissues during endosperm development [16]. However, the effects of LEA proteins on starch biosynthesis under high-temperature conditions remain unknown.

Here, we want to explore the quality factors influenced by environment and have identified an embryonic protein homologous to OsLEA3-2 [17], which we named OsLEA1b. The *oslea1b* mutants were more sensitive to high-temperature stress and drought stress than the wild type. Milled rice from *oslea1b* mutants showed significantly lower starch content and abnormal structure under a high-temperature environment, which resulted in significantly greater chalkiness and chalky grain rate than that from the wild type. A series of assays were utilized to determine the effects of OsLEA1b on starch biosynthesis under high-temperature conditions. These findings enhance our understanding of OsLEA proteins and grain filling in rice, providing useful information for genetic improvement of rice grain yield and quality.

## 2. Results

### 2.1. OsLEA1b Encodes an Embryonic Protein Mainly Expressed in the Endosperm

To isolate genes that regulate rice grain filling under high-temperature conditions, we searched the expression of genes that are induced under high temperatures in our RNA-Seq data (Appendix A). We identified *OsLEA1b* (LOC_Os01g16920), which is predominantly expressed during the grain-filling stage and is homologous to *OsLEA3-2* (Appendix A). Reverse transcription quantitative PCR (RT-qPCR) analysis validated that *OsLEA1b* is highly expressed in developing seeds, particularly during the late stages of seed development (Figure 1A). In contrast, *OsLEA1b* was rarely expressed in leaf and stem tissues. We transformed a vector harboring the *β-glucuronidase* (*GUS*) reporter gene driven by the *OsLEA1b* promoter into rice. Histochemical analysis corroborated that GUS activity in transgenic plants was strongest in the endosperm (Figure 1B), consistent with the RT-qPCR results. Transient expression analysis of a 35S:OsLEA1b–GFP construct in rice protoplasts revealed that the OsLEA1b–GFP fusion protein was widely present in the cytoplasm and nucleus, similar to the GFP control (Figure 1C). We therefore concluded that *OsLEA1b* encodes an embryonic protein.

### 2.2. The oslea1b Mutants Exhibit Chalky Endosperm and Decreased Yield

The *OsLEA1b* gene contains two exons and one intron and does not harbor any known domains aside from four low-complexity regions (determined using SMART, http://smart.embl-heidelberg.de, accessed on 1 January 2018, Appendix A). To investigate the function of *OsLEA1b* in rice endosperm development and plant growth, we knocked out *OsLEA1b* in the *japonica* rice cultivar ‘ZH11′ using clustered regularly interspaced short palindromic repeat (CRISPR)/CRISPR-associated nuclease 9 (Cas9)-mediated genome editing. We successfully obtained two independent homozygous mutants, *oslea1b-1* and *oslea1b-2*. RT-qPCR analysis showed that *OsLEA1b* expression in the two mutants was significantly lower than that in the wild type (Appendix A–C). Compared to the wild type, the grain width of *oslea1b* mutants was significantly lower, and this resulted in a lower 1000-grain weight even though the grains were longer in length (Figure 2A–C,H–K). Moreover, we observed that most mutant grains showed a semi-opaque endosperm, and the grain chalkiness rate in the mutants reached 95% (Figure 2D–G,O). A dramatically lower grain number per panicle in the mutants resulted in lower yields, only 49% relative to the wild-type control. But panicle length, tiller number, and seed setting rate were not significantly different in the mutants compared to the wild type (Figure 2L–N, Appendix A). Besides the shorter plant height, the grain-filling rate in the *oslea1b* mutants was slower on the 20th and 30th days after fertilization (DAF) than that of the wild type (Appendix A). Thus, the *OsLEA1b* gene plays an important role in grain quality and yield.

### 2.3. OsLEA1b Affects Starch Granule Development in the Endosperm

We next observed the starch grain morphology in endosperm cells of *oslea1b* mutants and the wild type. Scanning electron microscopy (SEM) revealed that wild-type starch granules were polygonal in shape and densely arranged. In contrast, starch granules in the chalky endosperm of the mutants were round and had numerous air spaces (Figure 3A–C). We prepared semi-thin sections of grains on the 10th DAF to observe the morphological differences in complex starch granules in the endosperm. Mutant grains had greater numbers of loosely arranged complex starch granules in peripheral and central endosperm cells than exhibited in wild-type grains (Figure 3D–G). In addition, gaps between amyloplasts in the center of the mutant endosperm were larger than those in wild-type endosperm, with many dispersed starch granules in the mutant (Figure 3H,I). Meanwhile, transmission electron microscopy (TEM) revealed many abnormal starch granules developed in the mutant endosperm (Figure 3J–O).

We examined additional physicochemical properties of starch from wild-type and *oslea1b* endosperm. Amylose, total starch, and protein contents in the mutants were significantly lower than those in the wild type (Figure 4A–C). The gel consistency of starch in the wild type was 81 mm, while that of *oslea1b* mutant starch was 39.5–53 mm (Figure 4D). The distribution of amylopectin chain lengths suggested a lower proportion of short chains with degrees of polymerization values between 6 and 13 but a higher proportion of intermediate chains (degrees 14 to 48) in the *oslea1b* mutants than that in the wild type (Figure 4E). Starch gelatinization and viscosity analyses indicated that the mutant starch was more difficult to gelatinize (Figure 4F,G). These results show that the physicochemical properties of endosperm starch were altered in the *oslea1b* mutants.

### 2.4. The oslea1b Mutants Showed Lower Grain Quality under High Temperature Conditions

High temperature during rice grain filling often reduces grain quality [18]. We grew wild-type and *oslea1b* plants under field conditions until flowering, after which plants were subjected to artificial high temperatures (35 °C for 12 h in light and 28 °C for 12 h in darkness) and normal temperatures (28 °C for 12 h in light and 22 °C for 12 h in darkness). Under high-temperature conditions, mature wild-type grains were slightly chalky, but mature *oslea1b* grains were floury with a significantly lower 1000-grain weight (Figure 5A–H). Compared to those under normal temperature conditions, wild-type and *oslea1b* grains had significantly lower amylose and total protein contents under high-temperature conditions. However, the differences between conditions were greater for *oslea1b* grains than for wild-type grains (Figure 5I–K).

We analyzed the expression of genes related to starch biosynthesis under high and normal temperatures. High temperature induced *OsLEA1b* expression at the early grain-filling stage (Appendix A). The expression levels of *STARCH BRANCHING ENZYME 1* (*OsBE1*) and *BRITTLE 1* (*OsBT1*) were significantly lower in the mutant than in the wild type under high temperature conditions. Conversely, the expression levels of *SOLUBLE STARCH SYNTHASE IIA* (*OsSSIIa*), *PULLULANASE* (*PUL*), *ISOAMYLASE 3* (*ISA3*), *STARCH BRANCHING ENZYME IIb* (*OsBEIIb*), *GRANULE-BOUND STARCH SYNTHASE I* (*GBSS1*), *RAG2*, and *GLUTELIN TYPE-D 1* (*GLUD1*) were significantly higher in the mutant than in the wild type under high temperatures (Figure 6). These differences in gene expression likely led to the differences observed in amylopectin’s fine structure in the mutants under high-temperature conditions.

### 2.5. OsLEA1b Promotes Drought Tolerance

The *oslea1b* mutants were sensitive to high temperature. We also examined the sensitivity of the *oslea1b* mutants to abiotic stresses such as drought and salinity. Treatment with 20% PEG6000 or 200 mM NaCl for 1 week induced greater reductions of root and shoot length in the mutant than in the wild type. Moreover, the survival rates of *oslea1b* mutants were only 13% and 11% when subjected to 20% PEG6000 (Figure 7A–C, Appendix A), indicating that the mutants were more sensitive to osmotic stress. When we exposed two-week-old seedlings to drought stress under soil conditions, the survival rate of *oslea1b* mutants was obviously reduced to 10% compared to that of the wild type (Figure 7D,E). *OsLEA1b* expression was induced after 12 h of 20% PEG6000 treatment in wild-type plants but not in the mutants (Figure 7F). These results indicate that the *oslea1b* mutants have reduced tolerance to drought stress. 

Reactive oxygen species (ROS) levels were determined in leaves of the wild type and the *oslea1b* mutants. There were no differences in various ROS indexes between control and stress treatments in the wild type. But *oslea1b* mutants had significantly increased peroxidase (POD) activity, hydrogen peroxide (H_2_O_2_) content, proline content, and malondialdehyde (MDA) content after PEG treatment compared to the wild type (Figure 7G–J). Genes related to stress, ROS scavenging, and ABA signaling were significantly up-regulated in the *oslea1b* mutants after PEG treatment but not in the control (Figure 7K,L). These results indicate that the *oslea1b* mutants are more sensitive to drought stress than the wild type.

## 3. Discussion

### 3.1. The oslea1b Mutants Showed Greater Sensitivity to High Temperature 

High temperature during the rice grain-filling stage often results in lower quantity and worse quality, such as higher chalkiness and lower grain-setting rate [19]. Grain chalkiness affects the appearance of rice. The imbalance of a source-to-sink under high temperature conditions leads to insufficient grain filling in endosperm cells and disordered enzyme activity, which bring a high grain chalkiness rate [20,21,22,23,24]. In this study, we observed that grains of *oslea1b* mutants exhibit high chalkiness rates under normal temperatures and chalky endosperm under high temperatures. The abnormally developed starch granules in the mutant endosperm cells were loosely arranged rather than whole complex granules. The *oslea1b* mutants were more sensitive to high temperature during endosperm development than wild-type plants. Moreover, the chalkiness rate and chalkiness degree of the mutants were dramatically higher than those of the wild type under both natural and artificial high temperatures. By contrast, the gelatinization and gel consistency of the two mutants were significantly lower than those in the wild type at high temperatures. Recent studies have shown that in addition to starch content, the arrangement of starch granules also affects rice grain cooking and eating qualities. It is likely that the abnormal starch granule development induced by high temperature in *oslea1b* mutants would affect the cooking and eating qualities. The higher proportion of short starch chain length and easier pasting properties of *oslea1b* may confer it greater sensitivity to high temperature tolerance.

Starch content and structure directly influence grain appearance such as chalkiness [25]. And many genes regulating starch biosynthesis were reported to influence rice chalkiness [26]. *Chalk5*, the major QTL control chalky grain rate of rice, two polymorphic nucleotides on the promoter of *Chalk5* affect its expression level. The lower expression level could decrease the chalkiness of rice and increase the percentage of milled rice. Total starch and amylose content in the *oslea1b* mutants were significantly lower than those in the wild type, especially under high temperatures. Moreover, a significantly different distribution of amylose chain lengths in the mutant led to an altered starch structure. The expression of *BE1*, *SSIIa*, *PUL*, *ISA3*, and *BEIIb* showed significant differences between the wild type and mutants under high-temperature treatment, which resulted in differences in the fine structure of starch [27,28]. The fine structure of starch affects starch grain expansion and gelatinization, which impacts the cooking performance and taste of rice [29]. Gelatinization temperature was higher in the mutants, causing hard, gluey consistency and reduced viscosity. The expression of *SSIIa* was significantly higher in the mutant than in the wild type under high temperatures, which may be responsible for the reduced viscosity. Although the effects of high temperature on rice quality are well known, additional research is needed to explore the underlying regulatory network to improve rice quality.

### 3.2. OsLEA1b Plays an Important Role in Rice Stress Responses 

Phylogenetic analysis showed that the embryonic protein OsLEA1b is homologous to the late embryogenesis abundant protein OsLEA3-2. LEA proteins are widely distributed in rice, Arabidopsis, maize (*Zea mays*), cotton (*Gossypium hirsutum*), potato (*Solanum tuberosum*), and other plant species [30,31,32,33,34]. The role of plant LEA proteins in stress tolerance has been well documented. Plants maintain cell membrane stability in low temperature environments through LEA protein biosynthesis [35]. In vitro and in vivo experiments demonstrated that soybean (*Glycine max*) PM2 (LEA3) protects proteins and maintains enzyme activity under unfavorable temperatures [36]. In addition, LEA proteins bind to certain ions to maintain osmotic pressure in cells. Two soybean LEA4 proteins, GmPM1 and GmPM9, possess metal-binding properties that reduce abiotic stress-induced oxidative damage [37]. 

Furthermore, LEA proteins enhance plant salt and drought tolerance. LEA proteins also play an important role in the adaptive responses to water deficit in higher plants [38]. Transgenic tobacco (*Nicotiana tabacum*) plants heterologously expressing melon (*Cucumis melo*) *CmLEA-S* display greater tolerance to drought and salt compared to wild-type plants [39]. In this study, *oslea1b* mutants were more sensitive to drought stress than wild-type plants, showing extremely low survival rates and significantly increased cellular ROS levels. *CATALASE B* (*OsCATB*), *ASCORBATE PEROXIDASE 1* (*OsAPX1*), *PEROXIDASE 1* (*OsPOX1*), *Ca^2+^/H^+^ EXCHANGER 1a* (*OsCAX1a*), *OsP5CR*, *OsNAC106*, *HIGH-AFFINITY K+ TRANSPORTER 1;3* (*OsHKT1;3*), *CYCLIC NUCLEOTIDE-GATED CHANNEL 9* (*OsCNGC9*), *RESPONSIVE TO ABA 21* (*OsRAB21*), *OsRAB16a*, and *AMMONIUM TRANSPORTER 3* (*OsAMT3*) are stress-related genes [40,41,42,43,44,45,46,47,48,49,50,51]. The stress-induced expression of these genes in *oslea1b* confers the sensitivity of the mutants to drought stress. This validates that *OsLEA1b* responds to drought stress by influencing the expression of those genes. Our results indicate that OsLEA1b regulates stress tolerance. We will further investigate the effects of overexpression of OsLEA1b on drought tolerance.

*OsLEA1b* is highly expressed in seeds yet barely detectable in other rice tissues. However, the drought sensitivity of *oslea1b* mutants was observed in other parts of the plant, such as the leaves. Additional studies are needed to explain how OsLEA1b, which is enriched in the endosperm, regulates plant stress tolerance. Overall, our research reveals that OsLEA1b regulates starch biosynthesis in the grain endosperm and plays an important role in rice drought tolerance. This study provides useful information for potential genetic improvement of rice yield and grain quality.

## 4. Materials and Methods

### 4.1. Plant Materials and Growth Conditions

The *oslea1b* mutants were generated by CRISPR/Cas9-mediated genome editing in the *japonica* rice (*Oryza sativa* ssp. *japonica*) of ‘Zhonghua 11′ (ZH11) background. Homozygous T_2_ lines of *oslea1b* were used in experiments. Rice plants were grown in a paddy field in Hangzhou, Zhejiang province, China, during the regular growing season. Wild-type and *oslea1b* plants (pre-flowering) were subjected to normal temperatures (12 h of light at 28 °C; 12 h of darkness at 22 °C) and high temperatures (12 h of light at 35 °C; 12 h of darkness at 28 °C) in plant growth chambers. The average outdoor daytime temperature in Hangzhou was 37 °C from 20 July 2020 to 20 August 2020.

### 4.2. Vector Construction and Plant Transformation

To knock out *OsLEA1b* using CRISPR/Cas9, single guide RNAs (sgRNA) targeting *OsLEA1b* (Os01g0276300) were cloned into the BGK03 vector (Biogle, Hangzhou, China; http://www.biogle.cn/index/excrispr, accessed on 1 October 2014). The CRISPR/Cas9–*OsLEA1b* construct was transformed into ZH11 calli via *Agrobacterium tumefaciens* (Agrobacterium)-mediated transformation. Primer sequences used for vector construction and validation are listed in Appendix A.

### 4.3. RNA Extraction, RT-qPCR Analysis, and GUS Staining

Total RNA was isolated from roots, stems, leaves, flowers, and developing grains at 5, 10, 15, 20, 25, and 30 days after fertilization (DAF) using an RNA extraction kit (Trizol, Invitrogen, Carlsbad, CA, USA) following the manufacturer’s instructions. Total RNA was reverse transcribed by priming with oligo (dT18) based on the instructions from the ReverTra Ace qPCR RT Kit (Toyobo, Osaka, Japan). Gene expression was quantified by reverse transcription quantitative PCR (RT-qPCR). RT-qPCR was performed using SYBR Green Real-time PCR Master Mix (Toyobo) and a Light Cycler 480 system (Roche, Basel, Switzerland). Relative gene expression was calculated using the 2^−ΔΔCT^ method [52]. Primer sequences used in this analysis are listed in Appendix A. The putative promoter region of *OsLEA1b* (~2 kb upstream of ATG) was amplified by PCR and cloned into the *Eco*RI and *Nco*I sites of pCAMBIA1305. This construct was transformed into ZH11 calli through Agrobacterium-mediated transformation. Positive T_0_ transgenic progeny were used to detect GUS activity as previously described [52].

### 4.4. Subcellular Localization

The *OsLEA1b* coding sequence was cloned into the pAN580-GFP vector capable of generating an OsLEA1b-GFP fusion protein. The fusion construct and an empty vector control (PAN580-GFP) were separately transformed into rice protoplasts [53]. GFP fluorescence was detected using a confocal laser scanning microscope with 488 nm (Leica TCS SP5, Wetzlar, Germany).

### 4.5. Electron Microscopy

Dry and completely brown rice grains of *oslea1b* mutants and the wild type were randomly selected. Images were obtained using a HITACHI S3400N scanning electron microscope (Hi-tachi, Tokyo, Japan). Scanning electron microscopy was performed as described previously [54]. To examine the development of compound granules, transverse sections (approximately 1-mm thick) of wild-type and *oslea1b* endosperms at 10 DAF were used to prepare semi-thin sections (800 nm). Samples were stained with I_2_-KI for 5 s and subsequently examined under a light microscope (Nikon Eclipse 80i, Tokyo, Japan; http://www.nikon.com accessed on 1 January 2018). All treatments were performed as described previously [55]. To observe amyloplast ultrastructure, developing seeds (10 DAF) of wild-type and *oslea1b* plants were analyzed using transmission electron microscopy (H7650; Hitachi). Samples were treated as described previously [56].

### 4.6. Analysis of Endosperm Starch Physicochemical Properties

Total starch and amylose contents were measured using the Megazyme K-TSTA and K-AMYL (Megazyme, Wicklow, Ireland) kits, respectively. Lipid and protein contents in *oslea1b* mutant and wild type were quantified following a previously established method [54]. Gel consistency was measured according to published methods [57]. To determine the pasting properties of endosperm starch, 3 g of milled rice flour (0.5 mm or less, 14% moisture) was transferred into a sample vessel containing 25 mL of distilled water. The sample was mixed and assessed using a Rapid Visco Analyzer (RVA Techmaster, Newport Scientific, Narrabeen, Australia). To determine the chain length distributions of amylopectin, 5 mg of rice powder was digested with *Pseudomonas amyloderamosa* isoamylase (Megazyme) and then analyzed by capillary electrophoresis (PA800 plus pharmaceutical analysis system, Beckman Coulter, Brea, CA, USA). 

### 4.7. Measurement of POD Activity and H_2_O_2_, Proline, and MDA Levels

Peroxidase activity in leaves was measured using an Amplex™ Red Hydrogen Peroxide/Peroxidase Assay Kit (A22180). MDA, proline, and H_2_O_2_ levels were determined by visible spectrophotometry using a malondialdehyde (MDA) assay kit (A003-1,TBA method), proline assay kit (A084-3-1), and hydrogen peroxide assay kit (A064-1-1, Nanjing, Jiancheng biotechnology company), respectively. 

### 4.8. Data Analysis

Data are presented as the means ± standard deviation (SD), shown by error bars. The chi-square (χ^2^) test and independent samples *t*-test (* *p* < 0.05; ** *p* < 0.01; NS, not significant) were used for statistical analysis using IBM SPSS Statistics 26 software.

## 5. Conclusions

In summary, we identified *OsLEA1b*, a heat-stress-responsive gene in rice grain filling. *OsLEA1b* is highly expressed in the endosperm, and its coding protein localizes to the nucleus and cytoplasm. The *oslea1b* mutants had abnormal compound starch granules in the endosperm cells and chalky endosperm with significantly decreased grain weight and grain number per panicle, as well as significantly lower contents of starch, protein, and lipid compared to the wild type. Moreover, *OsLEA1b* was induced by high temperature and drought stress. The *oslea1b* mutants showed more sensitivity to high temperatures. These results indicate that *OsLEA1b* regulates starch biosynthesis and influences rice grain quality, especially under high temperatures. In conclusion, this research provides a valuable gene resource and strategy for genetic improvement of rice grain quality.

## Figures and Tables

**Figure 1 plants-12-04070-f001:**
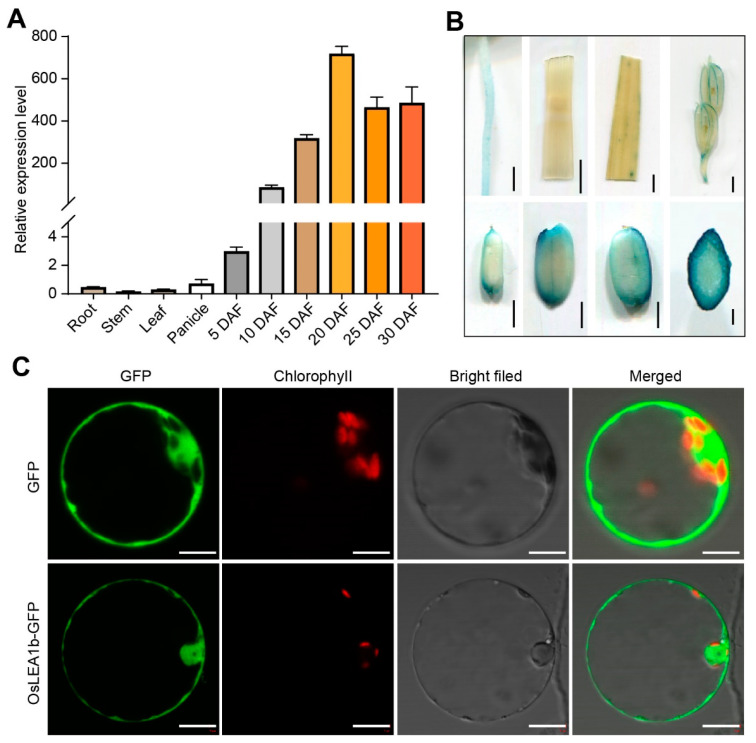
Expression pattern and subcellular localization of OsLEA1b. (**A**) Relative *OsLEA1b* expression levels in various tissues and in developing endosperms at 5, 10, 15, 20, 25, and 30 days after fertilization (DAF). Values are means ± SD from three biological replicates. (**B**) GUS staining in root, stem, leaf, spikelet, and developing grains expressing *ProOsLEA1b:GUS*. (**C**) Subcellular localization of OsLEA1b in rice protoplasts. Confocal microscopy images show OsLEA1b–GFP localized in the nucleus and cytoplasm. GFP signaling, chlorophyll autofluorescence, bright field, and merged images are shown for each construct. Scale bars, 2 mm in (**B**) and 10 μm in (**C**).

**Figure 2 plants-12-04070-f002:**
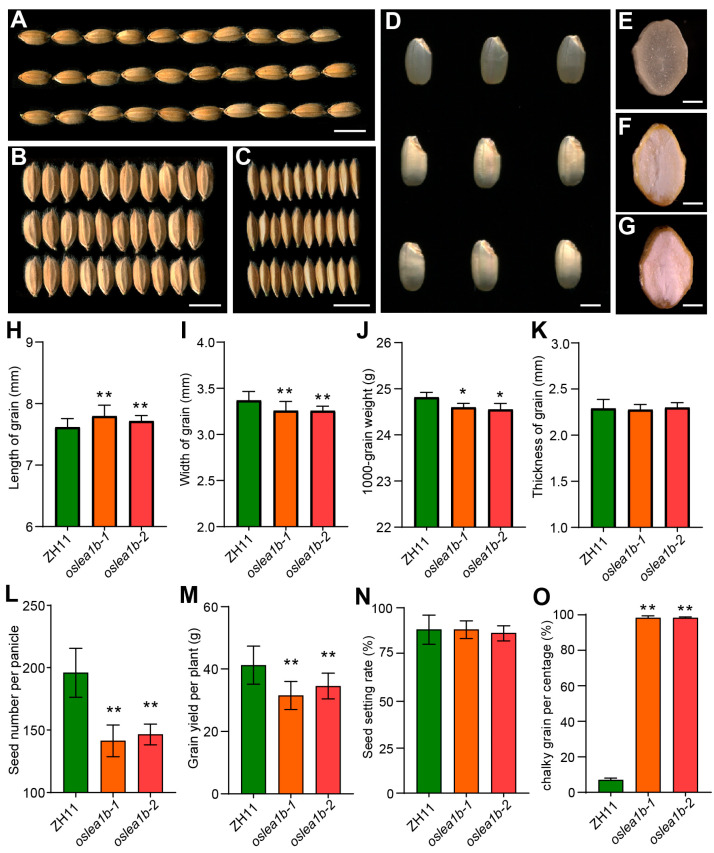
Analysis of agronomic traits of *oslea1b* mutants compared to the wild type. (**A**–**C**) Grain length, width, and thickness of wild-type (WT) and *oslea1b* grains. Scale bars, 6 mm. (**D**) Appearance of brown rice grains from WT and *oslea1b* mutant plants. Scale bar, 2 mm. (**E**–**G**) Transverse sections of WT (**E**) and *oslea1b* (**F**,**G**) endosperm. Scale bars, 2 mm. (**H**–**O**) Quantification of grain length (**H**), grain width (**I**), 1000-grain weight (**J**), grain thickness (**K**), seed number per panicle (**L**), grain yield per plant (**M**), seed setting rate (**N**), and chalky grain percentage (**O**) for WT plants and *oslea1b* mutants. Data are means ± SD from at least three biological replicates. Statistically significant differences were determined using Student’s *t*-test (*, *p* < 0.05, **, *p* < 0.01).

**Figure 3 plants-12-04070-f003:**
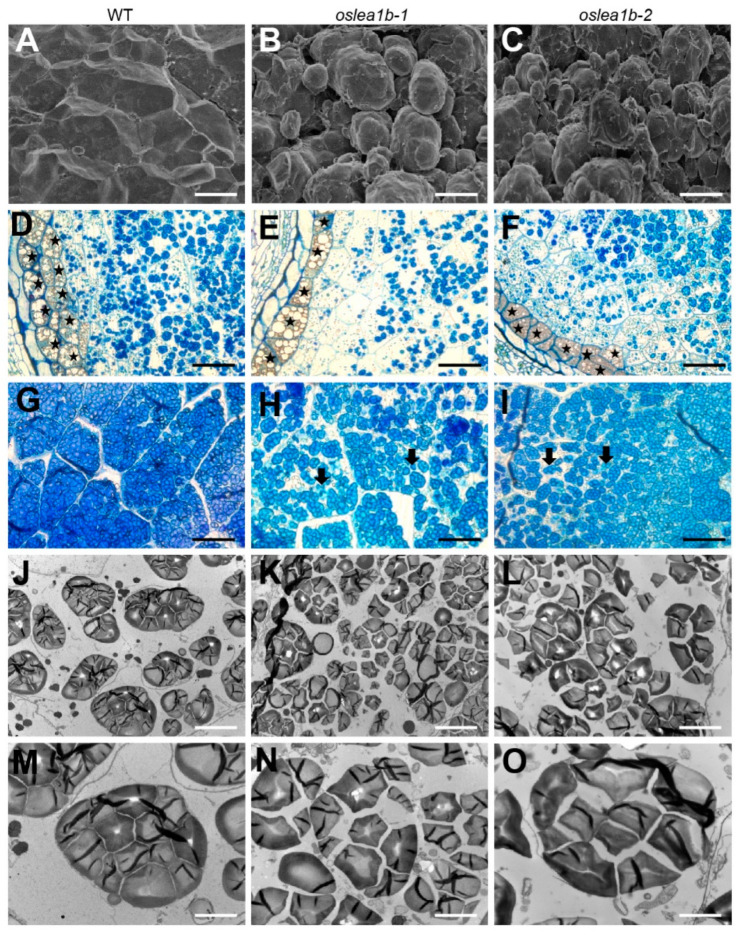
Starch granule formation and amyloplast development in endosperm cells of wild-type and *oslea1b* mutants. (**A**–**C**) Scanning electron microscopy of transverse sections of wild-type (**A**) and *oslea1b* (**B**,**C**) endosperm. (**D**–**I**) Semi-thin sections of WT and *oslea1b* endosperm at 10 days after fertilization (DAF). (**D**–**F**) Micrographs showing the peripheral region; (**G**–**I**) micrographs showing the central region. Black stars in (**D**–**F**) indicate aleurone cells. Arrows in (**H**,**I**) indicate broken amyloplasts. (**J**–**L**) Transmission electron microscopy shows well-developed amyloplasts in WT endosperm cells (**J**,**M**) and broken amyloplasts in *oslea1b* endosperm cells (**K**,**L**,**N**,**O**). Scale bars, 15 μm in (**A**–**C**), 40 μm in (**D**–**I**), 5 μm in (**J**–**L**), and 2 μm in (**M**–**O**).

**Figure 4 plants-12-04070-f004:**
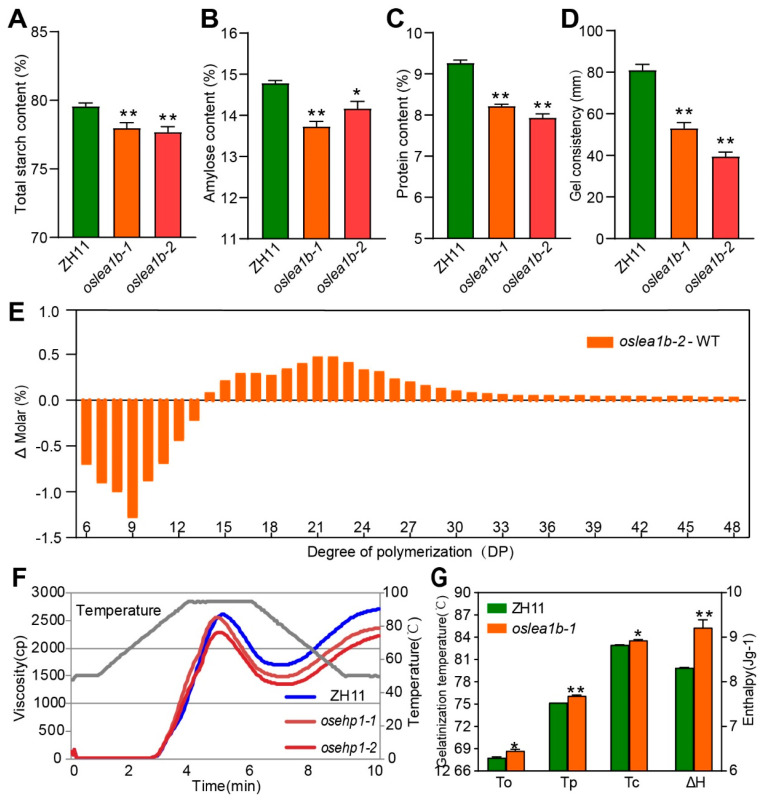
Starch physicochemical properties in the endosperm of *oslea1b* mutants. (**A**–**D**) Total starch content (**A**), amylose content (**B**), protein content (**C**), and gel consistency (**D**) of wild-type and *Osesp1* mutant endosperm. (**E**) Differences in amylopectin chain length distribution between WT and *oslea1b*. (**F**) Pasting properties of WT and *oslea1b* endosperm starch determined using a Rapid Visco Analyzer (RVA). The gray line indicates temperature changes during the measurements. (**G**) Gelatinization temperature of endosperm starch. To, Tp, and Tc represent the onset, peak, and concluding gelatinization temperatures, respectively. Data in (**A**–**D**,**G**) are presented as means ± SD from three replicates. Statistically significant differences were determined using Student’s *t*-test (*, *p* < 0.05, **, *p* < 0.01).

**Figure 5 plants-12-04070-f005:**
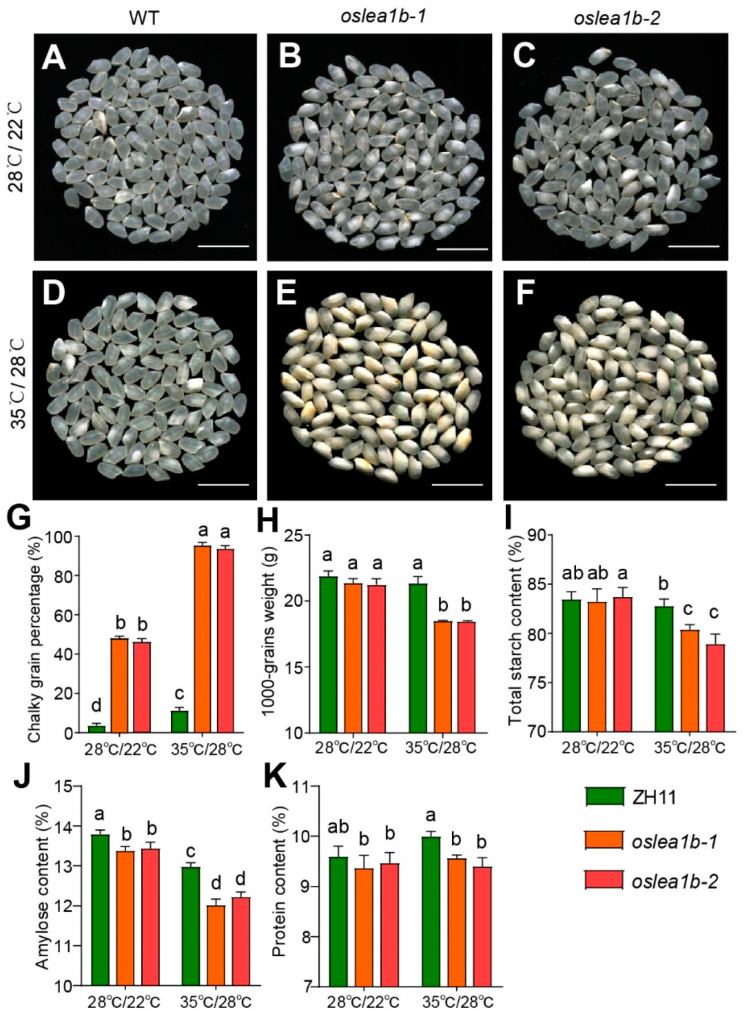
The *oslea1b* mutants were more sensitive to high temperatures during the grain-filling stage. (**A**–**F**) Appearance of wild-type (WT) and *oslea1b* mutant grains under high-temperature treatment (35 °C for 12 h in light and 28 °C for 12 h in darkness) and normal temperature treatment (28 °C for 12 h in light and 22 °C for 12 h in darkness). Scale bars, 1 cm. (**G**–**K**) Chalky grain percentage (**G**), 1000-grain weight (**H**), total starch content (**I**), amylose content (**J**), and total protein content (**K**) of WT and *oslea1b* mutant grains under normal and high temperatures. Different letters indicate significant differences at *p* < 0.05 by ANOVA and Duncan’s test.

**Figure 6 plants-12-04070-f006:**
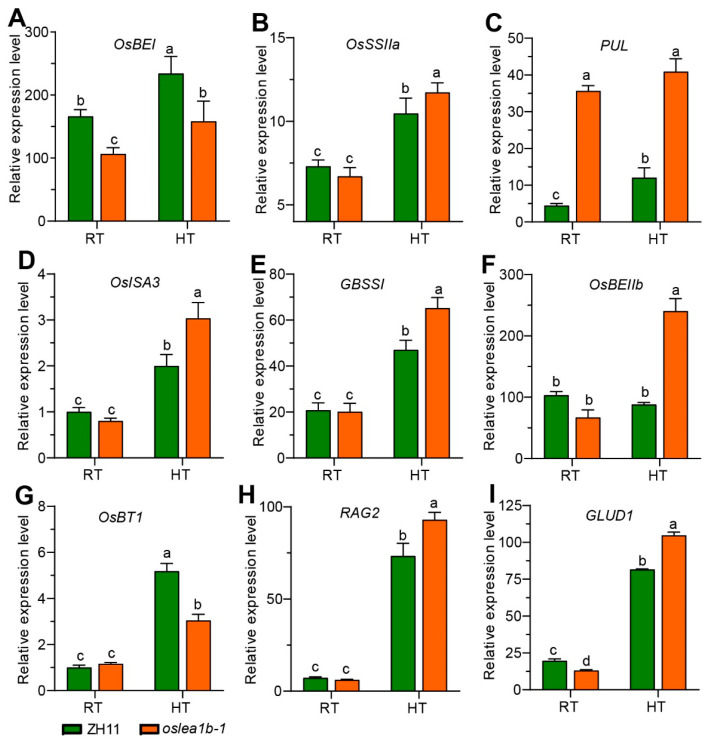
Transcription levels of genes related to regulate amylopectin synthesis in endosperm at 10 DAF. (**A**–**I**) Relative expression of genes related to amylopectin biosynthesis. Gene expression levels were measured in WT and *oslea1b-1* grains at 9 days after fertilization under high (HT) and normal (RT) temperature conditions. Different letters indicate significant differences at *p* < 0.05 by ANOVA and Duncan’s test.

**Figure 7 plants-12-04070-f007:**
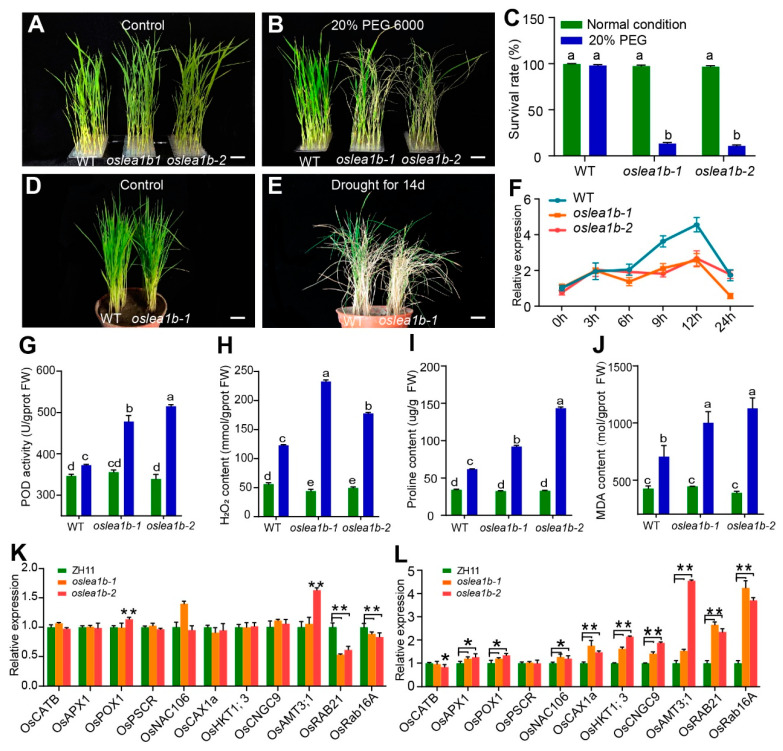
*OsLEA1b* gene positively regulates drought stress tolerance. (**A**,**B**) Phenotypes of seedlings of wild-type and *oslea1b* before and after 20% PEG treatment. (**C**) Survival rate of seedlings of WT and *oslea1b* after 20% PEG treatment. (**D**,**E**) Phenotypes of WT and *oslea1b* before and after drought stress. Water was withheld for 5 d followed by a 14-day recovery period. (**F**) Relative expression of *OsLEA1b* was induced by drought stress. (**G**–**J**) Peroxidase (POD) enzyme activity (**G**), and contents of hydrogen peroxide (H_2_O_2_; **H**), proline (**I**), and malondialdehyde (MDA; (**J)**) in WT and *oslea1b* mutant leaves. (**K**,**L**) Relative expression of genes related to reactive oxygen species scavenging and ABA signaling in wild type and mutants before (**K**) and after (**L**) 20% PEG treatment. Data are means ± SD from at least three biological replicates. Different letters in (**C**, **G–J**) indicate significant differences at *p* < 0.05 by ANOVA and Duncan’s test. In (**K**, **L**), statistically significant differences were determined using Student’s *t*-test (*, *p* < 0.05, **, *p* < 0.01).

## Data Availability

The datasets supporting the conclusions of this article are included within the article and Appendix A.

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
