# Peer review of "OsLEA1b Modulates Starch Biosynthesis at High Temperatures in Rice"

_plants, 2023, doi:10.3390/plants12234070_

Round 1
Reviewer 1 Report
Comments and Suggestions for Authors
The manuscript from Li et al describes the identification, expression, and consequences of mutation, of a protein they have named Endosperm Highly Expressed Protein 1 (EHP1) in rice. In particular, they have investigated the impact of its loss for starch synthesis, granule morphology, grain development and plant growth, as well as its induction during development and during abiotic stress. On the whole, the authors have provided a very detailed characterisation of the response of EHP1 to drought and temperature, as well as describing the impact of its knock-out (using CRISPR-Cas9).
My chief criticism is why give this protein a “new name”? The authors refer to the fact that it is a homologue of the LEA family of proteins and, as such, I strongly recommend they use a name and acronym which better reflects this high degree of homology. There is already enough confusion in the literature between the actual identities of proteins which are commonly referred to by their acronyms. The sequence they describe shows all the hallmarks of the LEA family, predicted to be a highly intrinsically disordered structure, with few introns in the gene, is induced by high temperature/drought, increases during development and grain filling, and possesses both a nuclear and cytosolic localisation. Further, the data are related to the transcription of mRNA for the protein, not the actual abundance of the protein itself, for which there are no data.
While the authors have amply described a number of important characteristics associated with the mutant phenotype, there is no clear indication as to the mechanism by which such effects are conferred. Very clearly, loss or expression of EHP1 is broad in impact, affecting not only expression of enzymes of starch synthesis but also anti-oxidants and ABA signalling and probably a wide variety of other processes. As a consequence, the Discussion tends to be repetitious of the Results, but this is perhaps inevitable given the nature of the investigation.
Overall, there is a great deal of useful characterisation of the expression of the LEA homologue and mutant phenotype, so the manuscript should be of wide interest and makes a very worthwhile contribution.
Comments on the Quality of English Language
A minor but significant point is that some attention needs to be paid to the use of English (which is otherwise commendable given that it is not the first language of the authors). For example:
Line 23: “induced” and “up-regulated” essentially mean the same thing, so just use one of the terms.
Line 70: “genes induced up-regulated”, change to “expression of genes which are induced”.
Line 163: “showed worser grain quality” change to “showed lower grain quality”.
Line 230: “showed more sensitive to high temperature” change to “showed greater sensitivity to high temperature”.
Line 231: “worser” change to “results in”.
Author Response
The manuscript from Li et al describes the identification, expression, and consequences of mutation, of a protein they have named Endosperm Highly Expressed Protein 1 (EHP1) in rice. In particular, they have investigated the impact of its loss for starch synthesis, granule morphology, grain development and plant growth, as well as its induction during development and during abiotic stress. On the whole, the authors have provided a very detailed characterization of the response of EHP1 to drought and temperature, as well as describing the impact of its knock-out (using CRISPR-Cas9).
Response: Thanks very much for your good comments.
My chief criticism is why give this protein a “new name”? The authors refer to the fact that it is a homologue of the LEA family of proteins and, as such, I strongly recommend they use a name and acronym which better reflects this high degree of homology. There is already enough confusion in the literature between the actual identities of proteins which are commonly referred to by their acronyms. The sequence they describe shows all the hallmarks of the LEA family, predicted to be a highly intrinsically disordered structure, with few introns in the gene, is induced by high temperature/drought, increases during development and grain filling, and possesses both a nuclear and cytosolic localization. Further, the data are related to the transcription of mRNA for the protein, not the actual abundance of the protein itself, for which there are no data.
Response: Thanks a lot for your constructive suggestion. After a deep reconsideration and research, we have renamed this protein as OsLEA1b. Because LOC_Os01g16920 is a homologue of the LEA family, it belongs to group 1 class of the LEA family and OsLEA1a have been reported (doi:10.1093/pcp/pcq172). As previous reports, the LEA proteins play an important role in drought tolerance, we have also evaluated the effects of OsLEA1b on stress tolerance and the results showed that the oslea1b mutants were sensitive to high temperature besides the drought stress. But unluckily we didn’t have the antibody for OsLEA1b, thus we couldn’t check the actual abundance of the protein itself. The whole manuscript have been revised. please check it.
While the authors have amply described a number of important characteristics associated with the mutant phenotype, there is no clear indication as to the mechanism by which such effects are conferred. Very clearly, loss or expression of EHP1 is broad in impact, affecting not only expression of enzymes of starch synthesis but also anti-oxidants and ABA signaling and probably a wide variety of other processes. As a consequence, the Discussion tends to be repetitious of the Results, but this is perhaps inevitable given the nature of the investigation.
Response: Thanks for your constructive suggestion. We have rediscussed the results. The higher proportion of short starch chain length and easier pasting properties of oslea1b may confer it greater sensitivity to high temperature tolerance. We will further investigate the effects of overexpression of OsLEA1b on drought tolerance.
Overall, there is a great deal of useful characterization of the expression of the LEA homologue and mutant phenotype, so the manuscript should be of wide interest and makes a very worthwhile contribution.
Response: Thanks very much for your comments and good suggestions, we have restructured the manuscript and modified the description of discussion after a deep reconsideration, please check it.
Comments on the Quality of English Language
A minor but significant point is that some attention needs to be paid to the use of English (which is otherwise commendable given that it is not the first language of the authors). For example:
Line 23: “induced” and “up-regulated” essentially mean the same thing, so just use one of the terms.
Line 70: “genes induced up-regulated”, change to “expression of genes which are induced”.
Response: Thank you for pointing them out. We have corrected it as you suggested.
Line 163: “showed worser grain quality” change to “showed lower grain quality”.
Response: Thank you for pointing it out. We have corrected it.
Line 230: “showed more sensitive to high temperature” change to “showed greater sensitivity to high temperature”.
Response: Thank you for pointing this out. We have corrected it as you suggested.
Line 231: “worser” change to “results in”.
Response: We very much appreciate your careful reading of our manuscript and pointing out these mistakes. Following your suggestions, all the mistakes in manuscript have been corrected, please check them in revised manuscript.
Reviewer 2 Report
Comments and Suggestions for Authors
Dear Authors,
Review of the manuscript titled: "ENDOSPERM HIGHLY EXPRESSED PROTEIN1 modulates starch biosynthesis at high temperature in rice" by Gang Li, Ruijie Cao, Liuyang Ma, Guiai Jiao, Pengfei Chen, Nannan Dong, Xinwei Li, Yingqing Duan, Xiaoxue Li, Mingdong Zhu, Gaoneng Shao, Zhonghua Sheng, Shikai Hu, Shaoqing Tang, Xiangjin Wei, Yinghong Yu, and Peisong Hu.
In this comprehensive study, Authors delve into the intricate mechanisms governing grain filling in rice under high-temperature stress, unveiling the pivotal role played by ENDOSPERM HIGHLY EXPRESSED PROTEIN1 (OsEHP1) in modulating starch biosynthesis and consequently impacting grain quality.
The researchers effectively identified OsEHP1 as a heat-stress-responsive gene predominantly expressed in the endosperm of rice. Through the analysis of knock-out mutants of OsEHP1, the team observed conspicuous abnormalities in compound starch granules within endosperm cells, resulting in the manifestation of chalky endosperm, accompanied by a significant reduction in both grain weight and the number of grains per panicle.
One of the noteworthy findings of this study was the altered proportion of short starch chains, specifically the decrease in chains with a degree from 6 to 12 and the simultaneous increase in chains of 12 to 48, in the osehp1 mutants as compared to the wild-type. This alteration corresponded to significantly lower levels of starch, protein, and lipid contents in the mutants, especially exacerbated under high-temperature conditions.
Furthermore, the Authors illuminated the role of OsEHP1 in responding to drought stress, noting its up-regulation under such conditions. Intriguingly, the study revealed a reduced survival rate of osehp1 mutants under drought stress, accompanied by a notable increase in reactive oxygen species levels, indicating the gene's involvement in stress response mechanisms.
The meticulous investigation conducted by the Authors provides compelling evidence supporting the pivotal regulatory role of OsEHP1 in orchestrating starch biosynthesis and influencing rice grain quality, particularly under high-temperature stress. The study not only sheds light on the molecular mechanisms underlying grain filling but also underscores the multifaceted functions of OsEHP1 in stress responses, establishing its significance in rice development and stress tolerance.
I believe that the research presented in the manuscript was conducted correctly, analyzed very thoroughly, and interestingly presented.
I only have a few minor comments that will help improve this manuscript:
1) Abstract - in this section I propose to provide a short research hypothesis and present the application nature of the obtained results.
2) Keywords - please do not use words in this section that are already in the title of the manuscript. Also, please arrange your keywords in alphabetical order.
3) Lines 50-66 - this is where the research hypothesis and the purpose of the work should be included. The research findings shouldn't be here.
4) Figure 2 - please explain what the dots on the figure mean.
5) Figures 5 and 6 - please explain what the letters above the bars mean. In addition, please provide information about +/- SD and the statistical test used.
6) Discussion - in this section, one could expect a slightly more thorough analysis of the results based on previous literature reports.
7) Conclusion - please provide the application of the results of your research.
8) The manuscript contains minor stylistic and linguistic errors. In addition, please adapt the manuscript more precisely to the requirements of the template used in the Palnts journal.
Overall, the manuscript presents a substantial contribution to the understanding of temperature-induced impacts on rice grain development and quality, offering valuable insights into potential targets for improving crop resilience and yield under adverse environmental conditions.
To sum up, I believe that the research is valuable and very well prepared for publication. Therefore, I encourage the Editors of the journal Palnts to consider publishing this manuscript.
Author Response
I believe that the research presented in the manuscript was conducted correctly, analyzed very thoroughly, and interestingly presented.
Response: Thanks very much for your professional and positive comments. It encourages us to do deeper research and investigation.
I only have a few minor comments that will help improve this manuscript:
- Abstract - in this section I propose to provide a short research hypothesis and present the application nature of the obtained results.
Response: Thanks a lot for your kind suggestions. We have added a short research hypothesis and application in the revised manuscript.
- Keywords - please do not use words in this section that are already in the title of the manuscript. Also, please arrange your keywords in alphabetical order.
Response: Thank you for your kind suggestions. We have renewed and rearranged the keywords. Please check it in the revised keywords.
- Lines 50-66 - this is where the research hypothesis and the purpose of the work should be included. The research findings shouldn't be here.
Response: Thank you for your good suggestions. In fact, the paragraph included line 55 to 65 was aimed to introduce the relationship between endosperm development and LEA proteins. And the effects of LEA proteins on starch biosynthesis under high-temperature conditions remained unknown. Then we have added the research hypothesis and the purpose of the work.
4) Figure 2 - please explain what the dots on the figure mean.
5) Figures 5 and 6 - please explain what the letters above the bars mean. In addition, please provide information about +/- SD and the statistical test used.
Response: Thank you for your kind suggestions. The dots on figure 2 mean individual sample. We have revised these above figure legends. Statistically significant differences were determined using Student’s t-test (indicated by different lowercase letters, P < 0.05). Please check in revised Figure 2, Figure 5 and Figure 6. The part of Data analysis had shown the information about +/- SD and the statistical test.
6) Discussion - in this section, one could expect a slightly more thorough analysis of the results based on previous literature reports.
Response: Many thanks for your suggestion, we have tried to deeply discuss the effects of OsLEA1b on the tolerance of high temperature and drought stress. The higher proportion of short starch chain length and easier pasting properties of oslea1b may confer it greater sensitivity to high temperature tolerance. We will further investigate the effects of overexpression of OsLEA1b on drought tolerance.
7) Conclusion - please provide the application of the results of your research.
Response: Thanks a lot for your constructive suggestion, we have provided the application of the research.
8) The manuscript contains minor stylistic and linguistic errors. In addition, please adapt the manuscript more precisely to the requirements of the template used in the Palnts journal.
Response: Thank you for your kind suggestions. We have revised the manuscript and modified the description of whole article after a deep check, please check it.
Overall, the manuscript presents a substantial contribution to the understanding of temperature-induced impacts on rice grain development and quality, offering valuable insights into potential targets for improving crop resilience and yield under adverse environmental conditions.
To sum up, I believe that the research is valuable and very well prepared for publication. Therefore, I encourage the Editors of the journal Palnts to consider publishing this manuscript.
Response: Thanks very much for your comments and good suggestions.